# An Information-Theoretic Analysis for Thompson Sampling with Many Actions

**Shi Dong**
Stanford University
sdong15@stanford.edu

**Benjamin Van Roy**
Stanford University
bvr@stanford.edu

## Abstract

Information-theoretic Bayesian regret bounds of Russo and Van Roy [8] capture the dependence of regret on prior uncertainty. However, this dependence is through entropy, which can become arbitrarily large as the number of actions increases. We establish new bounds that depend instead on a notion of rate-distortion. Among other things, this allows us to recover through information-theoretic arguments a near-optimal bound for the linear bandit. We also offer a bound for the logistic bandit that dramatically improves on the best previously available, though this bound depends on an information-theoretic statistic that we have only been able to quantify via computation.

## 1 Introduction

Thompson sampling [11] has proved to be an effective heuristic across a broad range of online decision problems [2, 10]. Russo and Van Roy [8] provided an information-theoretic analysis that yields insight into the algorithm's broad applicability and establishes a bound of $\sqrt{\overline{\Gamma} H(A^*) T}$ on cumulative expected regret over $T$ time periods of any algorithm and online decision problem. The *information ratio* $\overline{\Gamma}$ is a statistic that captures the manner in which an algorithm trades off between immediate reward and information acquisition; Russo and Van Roy [8] bound the information ratio of Thompson sampling for particular classes of problems. The entropy $H(A^*)$ of the optimal action quantifies the agent's initial uncertainty.

If the prior distribution of $A^*$ is uniform, the entropy $H(A^*)$ is the logarithm of the number of actions. As such, $\sqrt{\overline{\Gamma} H(A^*) T}$ grows arbitrarily large with the number of actions. On the other hand, even for problems with infinite action sets, like the linear bandit with a polytopic action set, Thompson sampling is known to obey gracious regret bounds [6]. This suggests that the dependence on entropy leaves room for improvement.

In this paper, we establish bounds that depend on a notion of rate-distortion instead of entropy. Our new line of analysis is inspired by rate-distortion theory, which is a branch of information theory that quantifies the amount of information required to learn an approximation [3]. This concept was also leveraged in recent work of Russo and Van Roy [9], which develops an alternative to Thompson sampling that aims to learn satisficing actions. An important difference is that the results of this paper apply to Thompson sampling itself.

We apply our analysis to linear and generalized linear bandits and establish Bayesian regret bounds that remain sharp with large action spaces. For the $d$-dimensional linear bandit setting, our bound is $O(d\sqrt{T \log T})$, which is tighter than the $O(d\sqrt{T} \log T)$ bound of [7]. Our bound also improves on the previous $O(\sqrt{dTH(A^*)})$ information-theoretic bound of [8] since it does not depend on

the number of actions. Our Bayesian regret bound is within a factor of $O(\sqrt{\log T})$ of the $\Omega(d\sqrt{T})$ worst-case regret lower bound of [4].

For the logistic bandit, previous bounds for Thompson sampling [7] and upper-confidence-bound algorithms [5] scale linearly with $\sup_x \phi'(x)/\inf_x \phi'(x)$, where $\phi$ is the logistic function $\phi(x) = e^{\beta x}/(1 + e^{\beta x})$. These bounds explode as $\beta \to \infty$ since $\lim_{\beta\to\infty} \sup_x \phi'(x) = \infty$. This does not make sense because, as $\beta$ grows, the reward of each action approaches a deterministic binary value, which should simplify learning. Our analysis addresses this gap in understanding by establishing a bound that decays as $\beta$ becomes large, converging to $2d\sqrt{T\log 3}$ for any fixed $T$. However, this analysis relies on a conjecture about the information ratio of Thompson sampling for the logistic bandit, which we only support through computational results.

## 2 Problem Formulation

We consider an online decision problem in which over each time period $t = 1, 2, \ldots$, an agent selects an action $A_t$ from a finite action set $\mathcal{A}$ and observes an outcome $Y_{A_t} \in \mathcal{Y}$, where $\mathcal{Y}$ denotes the set of possible outcomes. A fixed and known system function $g$ associates outcomes with actions according to

$$Y_a = g(a, \theta^*, W),$$

where $a \in \mathcal{A}$ is the action, $W$ is an exogenous noise term, and $\theta^*$ is the "true" model unknown to the agent. Here we adopt the Bayesian setting, in which $\theta^*$ is a random variable taking value in a space of parameters $\Theta$. The randomness of $\theta^*$ stems from the prior uncertainty of the agent. To make notations succinct and avoid measure-theoretic issues, we assume that $\Theta = \{\theta^1, \ldots, \theta^m\}$ is a finite set, whereas our analysis can be extended to the cases where both $\mathcal{A}$ and $\Theta$ are infinite.

The reward function $R : \mathcal{Y} \mapsto \mathbb{R}$ assigns a real-valued reward to each outcome. As a shorthand we define

$$\mu(a, \theta) = \mathbb{E}\left[R(Y_a)\big|\theta^* = \theta\right], \quad \forall a \in \mathcal{A}, \theta \in \Theta.$$

Simply stated, $\mu(a, \theta)$ is the expected reward of action $a$ when the true model is $\theta$. We assume that, conditioned on the true model parameter and the selected action, the reward is bounded[1], i.e.

$$\sup_{y\in\mathcal{Y}} R(y) - \inf_{y\in\mathcal{Y}} R(y) \le 1.$$

In addition, for each parameter $\theta$, let $\alpha(\theta)$ be the optimal action under model $\theta$, i.e.

$$\alpha(\theta) = \operatorname*{argmax}_{a\in\mathcal{A}} \mu(a, \theta).$$

Note that the ties induced by $\operatorname{argmax}$ can be circumvented by expanding $\Theta$ with identical elements. Let $A^* = \alpha(\theta^*)$ be the "true" optimal action and let $R^* = \mu(A^*, \theta^*)$ be the corresponding maximum reward.

Before making her decision at the beginning of period $t$, the agent has access to the *history* up to time $t - 1$, which we denote by

$$\mathcal{H}_{t-1} = \left(A_1, Y_{A_1}, \ldots, A_{t-1}, Y_{A_{t-1}}\right).$$

A *policy* $\pi = (\pi_1, \pi_2, \ldots)$ is defined as a sequence of functions mapping histories and exogenous noise to actions, which can be written as

$$A_t = \pi_t(\mathcal{H}_{t-1}, \xi_t), \quad t = 1, 2, \ldots,$$

where $\xi_t$ is a random variable which characterizes the algorithmic randomness. The performance of policy $\pi$ is evaluated by the finite horizon *Bayesian regret*, defined by

$$\text{BayesRegret}(T; \pi) = \mathbb{E}\left[\sum_{t=1}^{T}\left(R^* - R(Y_{A_t})\right)\right],$$

$$\mathbb{E}\left[\exp\left\{\lambda\left[R(g(a, \theta, W)) - \mu(a, \theta)\right]\right\}\right] \le \exp(\lambda^2/8) \quad \forall\lambda \in \mathbb{R}, a \in \mathcal{A}, \theta \in \Theta.$$

where the actions are chosen by policy $\pi$, and the expectation is taken over the randomness in both $R^*$ and $(A_t)_{t=1}^T$.

## 3 Thompson Sampling and Information Ratio

The Thompson sampling policy $\pi^{\mathrm{TS}}$ is defined such that at each period, the agent samples the next action according to her posterior belief of the optimal action, i.e.

$$\mathbb{P}\big(\pi_t^{\mathrm{TS}}(\mathcal{H}_{t-1}, \xi_t) = a \big| \mathcal{H}_{t-1}\big) = \mathbb{P}\big(A^* = a \big| \mathcal{H}_{t-1}\big), \quad \text{a.s. } \forall a \in \mathcal{A}, \ t = 1, 2, \dots.$$

An equivalent definition, which we use throughout our analysis, is that over period $t$ the agent samples a parameter $\theta_t$ from the posterior of the true parameter $\theta^*$, and plays the action $A_t = \alpha(\theta_t)$. The history available to the agent is thus

$$\tilde{\mathcal{H}}_t = \big(\theta_1, Y_{\alpha(\theta_1)}, \dots, \theta_t, Y_{\alpha(\theta_t)}\big).$$

The *information ratio*, first proposed in [8], quantifies the trade-off between exploration and exploitation. Here we adopt the simplified definition in [9], which integrates over all randomness. Let $\theta, \theta'$ be two $\Theta$-valued random variables. Over period $t$, the information ratio of $\theta'$ with respect to $\theta$ is defined by

$$\Gamma_t(\theta; \theta') = \frac{\mathbb{E}\big[R(Y_{\alpha(\theta)}) - R(Y_{\alpha(\theta')})\big]^2}{I\big(\theta; (\theta', Y_{\alpha(\theta')}) \big| \tilde{\mathcal{H}}_{t-1}\big)}, \tag{1}$$

where the denominator is the mutual information between $\theta$ and $(\theta', Y_{\alpha(\theta')})$, conditioned on the $\sigma$-algebra generated by $\tilde{\mathcal{H}}_{t-1}$. We can interpret $\theta$ as a benchmark model parameter that the agent wants to learn and $\theta'$ as the model parameter that she selects. When $\Gamma_t(\theta; \theta')$ is small, the agent would only incur large regret over period $t$ if she was expected to learn a lot of information about $\theta$. We restate a result proven in [6], which proposes a bound for the regret of any policy in terms of the worst-case information ratio.

**Proposition 1.** *For all $T > 0$ and policy $\pi$, let $(\theta_t)_{t=1}^T$ be such that $\alpha(\theta_t) = \pi_t(\mathcal{H}_{t-1}, \xi_t)$ for each $t = 1, 2 \dots, T$, then*

$$\mathrm{BayesRegret}(T; \pi) \leq \sqrt{\overline{\Gamma}_T \cdot H(\theta^*) \cdot T},$$

*where $H(\theta^*)$ is the entropy of $\theta^*$ and*

$$\overline{\Gamma}_T = \max_{1 \leq t \leq T} \Gamma_t(\theta^*; \theta_t).$$

The bound given by Proposition 1 is loose in the sense that it depends implicitly on the cardinality of $\Theta$. When $\Theta$ is large, knowing *exactly* what $\theta^*$ is requires a lot of information. Nevertheless, because of the correlation between actions, it suffices for the agent to learn a "blurry" version of $\theta^*$, which conveys far less information, to achieve low regret. In the following section we concretize this argument.

## 4 A Rate-Distortion Analysis of Thompson Sampling

In this section we develop a sharper bound for Thompson sampling. At a high level, the argument relies on the existence of a statistic $\psi$ of $\theta^*$ such that:

    i The statistic $\psi$ is less informative than $\theta^*$;

    ii In each period, if the agent aims to learn $\psi$ instead of $\theta^*$, the regret incurred can be bounded in terms of the information gained about $\psi$; we refer to this approximate learning as "compressed Thompson sampling";

    iii The regret of Thompson sampling is close to that of the compressed Thompson sampling based on the statistic $\psi$, and at the same time, compressed Thompson sampling yields no more information about $\psi$ than Thompson sampling.

Following the above line of analysis, we can bound the regret of Thompson sampling by the mutual information between the statistic $\psi$ and $\theta^*$. When $\psi$ can be chosen to be far less informative than $\theta^*$, we obtain a significantly tighter bound.

To develop the argument, we first quantify the amount of distortion that we incur if we replace one parameter with another. For two parameters $\theta, \theta' \in \Theta$, the distortion of $\theta$ with respect to $\theta'$ is defined as

$$d(\theta, \theta') = \mu(\alpha(\theta'), \theta') - \mu(\alpha(\theta), \theta'). \tag{2}$$

In other words, the distortion is the price we pay if we deem $\theta$ to be the true parameter while the actual true parameter is $\theta'$. Notice that from the definition of $\alpha$, we always have $d(\theta, \theta') \geq 0$. Let $\{\Theta_k\}_{k=1}^K$ be a partition of $\Theta$, i.e. $\bigcup_{k=1}^K \Theta_k = \Theta$ and $\Theta_i \cap \Theta_j = \emptyset$, $\forall i \neq j$, such that

$$d(\theta, \theta') \leq \epsilon, \quad \forall \theta, \theta' \in \Theta_k, \ k = 1, \ldots, K. \tag{3}$$

where $\epsilon > 0$ is a positive distortion tolerance. Let $\psi$ be the random variable taking values in $\{1, \ldots, K\}$ that records the index of the partition in which $\theta^*$ lies, i.e.

$$\psi = k \iff \theta^* \in \Theta_k. \tag{4}$$

Then we have $H(\psi) \leq \log K$. If the structure of $\Theta$ allows for a small number of partitions, $\psi$ would have much less information than $\theta^*$. Let subscript $t - 1$ denote corresponding values under the posterior measure $\mathbb{P}_{t-1}(\cdot) = \mathbb{P}(\cdot | \tilde{\mathcal{H}}_{t-1})$. In other words, $\mathbb{E}_{t-1}[\cdot]$ and $I_{t-1}(\cdot; \cdot)$ are random variables that are functions of $\tilde{\mathcal{H}}_{t-1}$. We claim the following.

**Proposition 2.** *Let $\psi$ be defined as in* (4). *For each $t = 1, 2, \ldots$, there exists a $\Theta$-valued random variable $\tilde{\theta}_t^*$ that satisfies the following:*

    *(i) $\tilde{\theta}_t^*$ is independent of $\theta^*$, conditioned on $\psi$.*

    *(ii) $\mathbb{E}_{t-1}\big[R^* - R(Y_{\alpha(\theta_t)})\big] - \mathbb{E}_{t-1}\big[R(Y_{\alpha(\tilde{\theta}_t^*)}) - R(Y_{\alpha(\tilde{\theta}_t)})\big] \leq \epsilon$, a.s.*

    *(iii) $I_{t-1}\big(\psi; (\tilde{\theta}_t, Y_{\alpha(\tilde{\theta}_t)})\big) \leq I_{t-1}\big(\psi; (\theta_t, Y_{\alpha(\theta_t)})\big)$, a.s.*

*where in (ii) and (iii), $\tilde{\theta}_t$ is independent from and distributed identically with $\tilde{\theta}_t^*$.*

According to Proposition 2, over period $t$ if the agent deviated from her original Thompson sampling scheme and applied a "one-step" compressed Thompson sampling to learn $\tilde{\theta}_t^*$ by sampling $\tilde{\theta}_t$, the extra regret that she would incur can be bounded (as is guaranteed by (ii)). Meanwhile, from (i), (iii) and the data-processing inequality, we have that

$$I_{t-1}\big(\tilde{\theta}_t^*; (\tilde{\theta}_t, Y_{\alpha(\tilde{\theta}_t)})\big) \leq I_{t-1}\big(\psi; (\tilde{\theta}_t, Y_{\alpha(\tilde{\theta}_t)})\big) \leq I_{t-1}\big(\psi; (\theta_t, Y_{\alpha(\theta_t)})\big), \text{ a.s.} \tag{5}$$

which implies that the information gain of the compressed Thompson sampling will not exceed that of the original Thompson sampling towards $\psi$. Therefore, the regret of the original Thompson sampling can be bounded in terms of the total information gain towards $\psi$ and the worst-case information ratio of the one-step compressed Thompson sampling. Formally, we have the following.

**Theorem 1.** *Let $\{\Theta_k\}_{k=1}^K$ be any partition of $\Theta$ such that for any $k = 1, \ldots, K$ and $\theta, \theta' \in \Theta_k$, $d(\theta, \theta') \leq \epsilon$. Let $\psi$ be defined as in* (4) *and let $\tilde{\theta}_t^*$ and $\tilde{\theta}_t$ satisfy the conditions in Proposition 2. We have*

$$\text{BayesRegret}(T; \pi^{\text{TS}}) \leq \sqrt{\bar{\Gamma} \cdot I(\theta^*; \psi) \cdot T} + \epsilon \cdot T, \tag{6}$$

*where*

$$\bar{\Gamma} = \max_{1 \leq t \leq T} \Gamma_t(\tilde{\theta}_t^*; \tilde{\theta}_t).$$

**Proof.** We have that

$$
\begin{aligned}
\mathrm{BayesRegret}(T; \pi^{\mathrm{TS}}) &= \sum_{t=1}^{T} \mathbb{E}\Big[ R^* - R(Y_{A_t}) \Big] \\
&= \sum_{t=1}^{T} \mathbb{E}\Big\{ \mathbb{E}_{t-1}\Big[ R^* - R(Y_{A_t}) \Big] \Big\} \\
&\overset{(a)}{\leq} \sum_{t=1}^{T} \mathbb{E}\Big\{ \mathbb{E}_{t-1}\Big[ R(Y_{\alpha(\tilde{\theta}_t^*)}) - R(Y_{\alpha(\tilde{\theta}_t)}) \Big] \Big\} + \epsilon \cdot T \\
&= \sum_{t=1}^{T} \sqrt{ \Gamma_t(\tilde{\theta}_t^*, \tilde{\theta}_t) \cdot I\Big( \tilde{\theta}_t^*; (\tilde{\theta}_t, Y_{\alpha(\tilde{\theta}_t)}) \big| \tilde{\mathcal{H}}_{t-1} \Big) } + \epsilon \cdot T \\
&\overset{(b)}{\leq} \sum_{t=1}^{T} \sqrt{ \overline{\Gamma} \cdot I\Big( \psi; (\theta_t, Y_{\alpha(\theta_t)}) \big| \tilde{\mathcal{H}}_{t-1} \Big) } + \epsilon \cdot T \\
&\overset{(c)}{\leq} \sqrt{ \overline{\Gamma} \cdot T \cdot \sum_{t=1}^{T} I\Big( \psi; (\theta_t, Y_{\alpha(\theta_t)}) \big| \tilde{\mathcal{H}}_{t-1} \Big) } + \epsilon \cdot T \\
&\overset{(d)}{=} \sqrt{ \overline{\Gamma} \cdot T \cdot I\Big( \psi; \tilde{\mathcal{H}}_{T-1} \Big) } + \epsilon \cdot T \\
&\overset{(e)}{\leq} \sqrt{ \overline{\Gamma} \cdot T \cdot I\Big( \theta^*; \psi \Big) } + \epsilon \cdot T,
\end{aligned}
\tag{7}
$$

where $(a)$ follows from Proposition 2 (ii); $(b)$ follows from (5); $(c)$ results from Cauchy-Schwartz inequality; $(d)$ is the chain rule for mutual information and $(e)$ comes from that

$$
I\Big( \psi; \tilde{\mathcal{H}}_T \Big) \leq I\Big( \psi; (\theta^*, \tilde{\mathcal{H}}_T) \Big) = I\Big( \psi; \theta^* \Big) + I\Big( \psi; \tilde{\mathcal{H}}_T \big| \theta^* \Big) = I\Big( \psi; \theta^* \Big),
$$

where we use the fact that $\psi$ is independent of $\tilde{\mathcal{H}}_T$, conditioned on $\theta^*$. Thence we arrive at our desired result. $\qquad \square$

**Remark.** The bound given in Theorem 1 dramatically improves the previous bound in Proposition 1 since $I(\theta^*; \psi)$ can be bounded by $H(\psi)$, which, when $\Theta$ is large, can be much smaller than $H(\theta^*)$. The new bound also characterizes the tradeoff between the preserved information $I(\theta^*; \psi)$ and the distortion tolerance $\epsilon$, which is the essence of rate distortion theory. In fact, we can define the distortion between $\theta^*$ and $\psi$ as

$$
D(\theta^*, \psi) = \max_{1 \leq t \leq T} \mathrm{esssup}\, \Big\{ \mathbb{E}_{t-1}\big[ R^* - R(Y_{\alpha(\theta_t)}) \big] - \mathbb{E}_{t-1}\big[ R(Y_{\alpha(\tilde{\theta}_t^*)}) - R(Y_{\alpha(\tilde{\theta}_t)}) \big] \Big\},
$$

where $\tilde{\theta}_t^*$ and $\tilde{\theta}_t$ depend on $\psi$ through Proposition 2. By taking the infimum over all possible choices of $\psi$, the bound (6) can be written as

$$
\mathrm{BayesRegret}(T; \pi^{\mathrm{TS}}) \leq \sqrt{ \overline{\Gamma} \cdot \rho(\epsilon) \cdot T } + \epsilon \cdot T, \quad \forall \epsilon > 0,
\tag{8}
$$

where

$$
\begin{aligned}
\rho(\epsilon) = \quad &\min_{\psi} \quad I(\theta^*; \psi) \\
&\text{s.t.} \quad D(\theta^*, \psi) \leq \epsilon
\end{aligned}
$$

is the *rate-distortion function* with respect to the distortion $D$.

To obtain explicit bounds for specific problem instances, we use the fact that $I(\theta^*; \psi) \leq H(\psi) \leq \log K$. In the following section we introduce a broad range of problems in which both $K$ and $\overline{\Gamma}$ can be effectively bounded.

## 5 Specializing to Structured Bandit Problems

We now apply the analysis in Section 2 to common bandit settings and show that our bounds are significantly sharper than the previous bounds. In these models, the observation of the agent is the received reward. Hence we can let $R$ be the identity function and use $R_a$ as a shorthand for $R(Y_a)$.

## 5.1 Linear Bandits

Linear bandits are a class of problems in which each action is parametrized by a finite-dimensional feature vector, and the mean reward of playing each action is the inner product between the feature vector and the model parameter vector. Formally, let $\mathcal{A}, \Theta \subset \mathbb{R}^d$, where $d < \infty$, and $\mathcal{Y} \subseteq [-1/2, 1/2]$. The reward of playing action $a$ satisfies

$$\mathbb{E}[R_a | \theta^* = \theta] = \mu(a, \theta) = \frac{1}{2} a^\top \theta, \quad \forall a \in \mathcal{A}, \theta \in \Theta.$$

Note that we apply a normalizing factor $1/2$ to make the setting consistent with our assumption that $\sup_y R(y) - \inf_y R(y) \le 1$.

A similar line of analysis as in [8] allows us to bound the information ratio of the one-step compressed Thompson sampling.

**Proposition 3.** *Under the linear bandit setting, for each $t = 1, 2, \ldots$, letting $\tilde{\theta}_t^*$ and $\tilde{\theta}_t$ satisfy the conditions in Proposition 2, we have*

$$\Gamma_t(\tilde{\theta}_t^*; \tilde{\theta}_t) \le \frac{d}{2}.$$

At the same time, with the help of a covering argument, we can also bound the number of partitions that is required to achieve distortion tolerance $\epsilon$.

**Proposition 4.** *Under the linear bandit setting, suppose that $\mathcal{A}, \Theta \subseteq \overline{\mathbf{B}_d(0, 1)}$, where $\overline{\mathbf{B}_d(0, 1)}$ is the d-dimensional closed Euclidean unit ball. Then for any $\epsilon > 0$ there exists a partition $\{\Theta_k\}_{k=1}^K$ of $\Theta$ such that for all $k = 1, \ldots, K$ and $\theta, \theta' \in \Theta_k$, we have $d(\theta, \theta') \le \epsilon$ and*

$$K \le \left( \frac{1}{\epsilon} + 1 \right)^d.$$

Combining Theorem 1, Propositions 3 and 4, we arrive at the following bound.

**Theorem 2.** *Under the linear bandit setting, if $\mathcal{A}, \Theta \subseteq \overline{\mathbf{B}_d(0, 1)}$, then*

$$\mathrm{BayesRegret}(T; \pi^{\mathrm{TS}}) \le d \sqrt{T \log \left( 3 + \frac{3\sqrt{2T}}{d} \right)}.$$

This bound is the first information-theoretic bound that does not depend on the number of available actions. It significantly improves the bound $O\left( \sqrt{dT \cdot H(A^*)} \right)$ in [8] and the bound $O\left( \sqrt{|\mathcal{A}|T \log |\mathcal{A}|} \right)$ in [1] in that it drops the dependence on the cardinality of the action set and imposes no assumption on the reward distribution. Comparing with the confidence-set-based analysis in [7], which results in the bound $O(d\sqrt{T} \log T)$, our argument is much simpler and cleaner and yields a tighter bound. This bound suggests that Thompson sampling is near-optimal in this context since it exceeds the minimax lower bound $\Omega(d\sqrt{T})$ proposed in [4] by only a $\sqrt{\log T}$ factor.

## 5.2 Generalized Linear Bandits with iid Noise

In generalized linear models, there is a fixed and strictly increasing *link function* $\phi : \mathbb{R} \mapsto [0, 1]$, such that

$$\mathbb{E}[R_a | \theta^* = \theta] = \mu(a, \theta) = \phi(a^\top \theta).$$

Let

$$\underline{L} = \inf_{a \in \mathcal{A}, \theta \in \Theta} a^\top \theta, \quad \overline{L} = \sup_{a \in \mathcal{A}, \theta \in \Theta} a^\top \theta.$$

We make the following assumptions.

**Assumption 1.** *The reward noise is iid, i.e.*

$$R_a = \mu(a, \theta^*) + W_a = \phi(a^\top \theta^*) + W_a, \quad \forall a \in \mathcal{A},$$

*where $W_a$ is a zero-mean noise term with a fixed and known distribution for all $a \in \mathcal{A}$.*

**Assumption 2.** *The link function $\phi$ is continuously differentiable in $[\underline{L}, \overline{L}]$, with*

$$C(\phi) = \sup_{x \in [\underline{L}, \overline{L}]} \phi'(x) < \infty.$$

Under these assumptions, both the information ratio of the compressed Thompson sampling and the number of partitions can be bounded.

**Proposition 5.** *Under the genearlized linear bandit setting and Assumptions 1 and 2, for each $t = 1, 2, \ldots$, letting $\tilde{\theta}_t^*$ and $\tilde{\theta}_t$ satisfy the conditions in Proposition 2, we have*

$$\Gamma_t(\tilde{\theta}_t^*; \tilde{\theta}_t) \leq 2C(\phi)^2 d.$$

**Proposition 6.** *Under the generalized linear bandit setting and Assumption 2, suppose that $\mathcal{A}, \Theta \subseteq \overline{\mathbf{B}}_d(0, 1)$. Then for any $\epsilon > 0$ there exists a partition $\{\Theta_k\}_{k=1}^K$ of $\Theta$ such that for each $k = 1, \ldots, K$ and $\theta, \theta' \in \Theta_k$ we have $d(\theta, \theta') \leq \epsilon$ and*

$$K \leq \left( \frac{2C(\phi)}{\epsilon} + 1 \right)^d.$$

Combining Theorem 1, Propositions 5 and 6, we have the following.

**Theorem 3.** *Under the generalized linear bandit setting and Assumptions 1 and 2, if $\mathcal{A}, \Theta \subseteq \overline{\mathbf{B}}_d(0, 1)$, then*

$$\text{BayesRegret}(T; \pi^{\text{TS}}) \leq 2C(\phi) \cdot d \sqrt{T \log \left( 3 + \frac{3\sqrt{2T}}{d} \right)}.$$

Note that the optimism-based algorithm in [5] achieves $O(rd\sqrt{T} \log T)$ regret, and the bound of Thompson sampling given in [7] is $O(rd\sqrt{T} \log^{3/2} T)$, where $r = \sup_x \phi'(x) / \inf_x \phi'(x)$. Theorem 3 apparently yields a sharper bound.

### 5.3 Logistic Bandits

Logistic bandits are special cases of generalized linear bandits, in which the agent only observes binary rewards, i.e. $\mathcal{Y} = \{0, 1\}$. The link function is given by $\phi^{\text{L}}(x) = e^{\beta x}/(1 + e^{\beta x})$, where $\beta \in (0, \infty)$ is a fixed and known parameter. Conditioned on $\theta^* = \theta$, the reward of playing action $a$ is Bernoulli distributed with parameter $\phi^{\text{L}}(a^\top \theta)$.

The preexisting upper bounds on logistic bandit problems all scale linearly with

$$r = \sup_x (\phi^{\text{L}})'(x) / \inf_x (\phi^{\text{L}})'(x),$$

which explodes when $\beta \to \infty$. However, when $\beta$ is large, the rewards of actions are clearly bifurcated by a hyperplane and we expect Thompson sampling to perform better. The regret bound given by our analysis addresses this point and has a finite limit as $\beta$ increases. Since the logistic bandit setting is incompatible with Assumption 1, we propose the following conjecture, which is supported with numerical evidence.

**Conjecture 1.** *Under the logistic bandit setting, let the link function be $\phi^{\text{L}}(x) = e^{\beta x}/(1 + e^{\beta x})$, and for each $t = 1, 2 \ldots$, let $\tilde{\theta}_t^*$ and $\tilde{\theta}_t$ satisfy the conditions in Proposition 2. Then for all $\beta \in (0, \infty)$,*

$$\Gamma_t(\tilde{\theta}_t^*; \tilde{\theta}_t) \leq \frac{d}{2}.$$

To provide evidence for Conjecture 1, for each $\beta$ and $d$, we randomly generate 100 actions and parameters and compute the exact information ratio under a randomly selected distribution over the parameters. The result is given in Figure 1. As the figure shows, the simulated information ratio is always smaller than the conjectured upper bound $d/2$. We suspect that for every link function $\phi$, there exists an upper bound for the information ratio that depends only on $d$ and $\phi$ and is independent of the cardinality of the parameter space. This opens an interesting topic for future research.

We further make the following assumption, which posits existence of a classification margin that applies uniformly over $\theta \in \Theta$

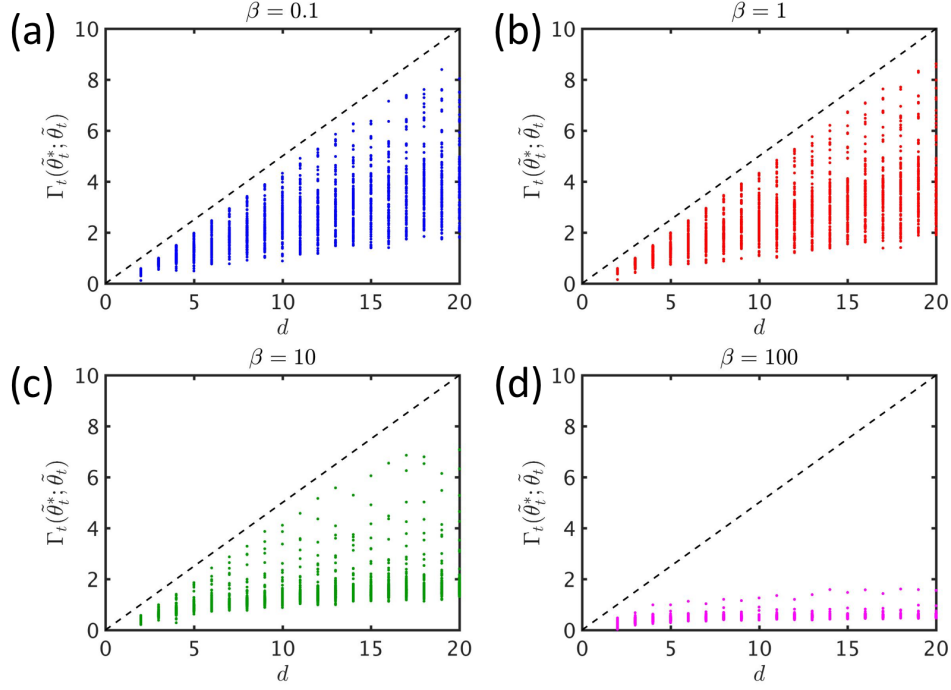

Figure 1: Simulated information ratio values for dimensions $d = 2, 3, \ldots, 20$ and (a) $\beta = 0.1$, (b) $\beta = 1$, (c) $\beta = 10$ and (d) $\beta = 100$. The diagonal black dashed line is the upper bound $\Gamma = d/2$.

**Assumption 3.** *We have that* $\inf_{\theta \in \Theta} |\mu(\alpha(\theta), \theta) - 1/2| > 0$. *Equivalently, we have that*

$$\inf_{\theta \in \Theta} \left| \alpha(\theta)^\top \theta \right| > 0.$$

The following theorem introduces the bound for the logistic bandit.

**Theorem 4.** *Under the logistic bandit setting where* $\mathcal{A}, \Theta \subseteq \overline{\mathbf{B}_d(0,1)}$, *for all* $\beta > 0$, *if the link function is given by* $\phi^{\mathrm{L}}(x) = e^{\beta x}/(1 + e^{\beta x})$, *Assumption 3 holds with* $\inf_{\theta \in \Theta} \left| \alpha(\theta)^\top \theta \right| = \delta > 0$, *and Conjecture 1 holds, then for all sufficiently large* $T$,

$$\mathrm{BayesRegret}(T; \pi^{\mathrm{TS}}) \leq 2d \sqrt{T \log \left( 3 + \frac{6\sqrt{2T}}{d} \cdot \frac{\beta e^{\beta \delta}}{(1 + e^{\beta \delta})^2} \right)} \tag{9}$$

$$\leq 2d \sqrt{T \log \left( 3 + \frac{3\sqrt{2T}}{2d} \cdot \min \left\{ \delta^{-1}, \beta \right\} \right)}. \tag{10}$$

For fixed $d$ and $T$, when $\beta \to \infty$ the right-hand side of (9) converges to $2d\sqrt{T \log 3}$. Thus (9) is substantially sharper than previous bounds when $\beta$ is large.

## 6 Conclusion

Through an analysis based on rate-distortion, we established a new information-theoretic regret bound for Thompson sampling that scales gracefully to large action spaces. Our analysis yields an $O(d\sqrt{T \log T})$ regret bound for the linear bandit problem, which strengthens state-of-the-art bounds. The same regret bound applies also to the logistic bandit problem if a conjecture about the information ratio that agrees with computational results holds. We expect that our new line of analysis applies to a wide range of online decision algorithms.

**Acknowledgments**

This work was supported by a grant from the Boeing Corporation and the Herb and Jane Dwight Stanford Graduate Fellowship. We would also like to thank Daniel Russo, David Tse and Xiuyuan Lu for useful conversations.

## Footnotes

[1]The boundedness assumption allows application of a basic version of Pinsker's inequality. Since there exists a version of Pinsker's inequality that applies to sub-Gaussian random variables (see Lemma 3 of [8]), all of our results hold without change for $1/4$-sub-Gaussian rewards, i.e.

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
