[Supplementary Material]

# A  Proof of Proposition 2

We first show the following lemma.

**Lemma 1.** *Let $\{a_i\}_{i=1}^N$ and $\{b_i\}_{i=1}^N$ be two sequences of real numbers, where $N < \infty$. Let $\{p_i\}_{i=1}^N$ be such that $p_i \geq 0$ for all $i$ and $\sum_{m=1}^N p_m = 1$. Then there exist indices $j, k \in \{1, \cdots, N\}$ (possibly $j = k$) and $p \in [0, 1]$ such that*

$$pa_j + (1-p)a_k \leq \sum_{m=1}^N a_m p_m$$

*and*

$$pb_j + (1-p)b_k \leq \sum_{m=1}^N b_m p_m.$$

**Proof.** We prove the lemma by induction over $N$. The result is trivial when $N = 1, 2$. Assume that the result holds when $N = n$. In the following we show the case where $N = n + 1$. Let $A = \sum_{m=1}^{n+1} a_m p_m$ and $B = \sum_{m=1}^{n+1} b_m p_m$.

Suppose there exists index $t \in \{1, \cdots, n+1\}$ such that $a_t \leq A$ and $b_t \leq B$, then by choosing $j = k = t$, there is

$$pa_j + (1-p)a_k = a_t \leq A \quad \text{and} \quad pb_j + (1-p)b_k = b_t \leq B.$$

Suppose there exists index $t \in \{1, \cdots, n+1\}$ such that $a_t \geq A$ and $b_t \geq B$. Without loss of generality we can assume $t = n + 1$. If $p_{n+1} = 1$, the result becomes trivial by choosing $j = k = n + 1$. Hence we only consider $p_{n+1} < 1$. Let $p'_i = p_i/(1 - p_{n+1})$ for $i = 1, \cdots, n$, then $\sum_{m=1}^n p'_m = 1$. Applying our assumption to $\{a_i\}_{i=1}^n$, $\{b_i\}_{i=1}^n$ and $\{p'_i\}_{i=1}^n$, we can find $j', k' \in \{1, \cdots, n\}$ and $p' \in [0, 1]$ such that

$$p'a_{j'} + (1-p')a_{k'} \leq \sum_{m=1}^n a_m p'_m$$

and

$$p'b_{j'} + (1-p')b_{k'} \leq \sum_{m=1}^n b_m p'_m.$$

Notice that

$$\sum_{m=1}^n a_m p'_m = \sum_{m=1}^n \frac{a_m p_m}{1 - p_{n+1}} \leq \sum_{m=1}^{n+1} a_m p_m = A,$$

and similarly $\sum_{m=1}^n b_m p'_m \leq B$. Therefore by choosing $j = j', k = k'$ and $p = p'$, we arrive at the result.

Consequently, we only have to consider the case where for each $t \in \{1, \cdots, n+1\}$, either $a_t > A, b_t < B$ or $a_t < A, b_t > B$. Without loss of generality, let $s$ be the index such that $a_t > A, b_t < B, \forall t \in \{1, \cdots, s\}$ and $a_t < A, b_t > B, \forall t \in \{s+1, \cdots, n+1\}$. Suppose the result is false, then for any $\ell \in \{1, \cdots, s\}$ and $h \in \{s+1, \cdots, n+1\}$, the following set of inequalities

$$\begin{cases} pa_\ell + (1-p)a_h \leq A \\ pb_\ell + (1-p)b_h \leq B \end{cases}$$

has no solution for $p$. Since $a_\ell > A > a_h$ and $b_\ell < B < b_h$, this can only happen when

$$\frac{A - a_h}{a_\ell - a_h} < \frac{b_h - B}{b_h - b_\ell}.$$

Rearranging, the above inequality is equivalent to

$$b_h A - b_\ell A + a_\ell B - a_h B + a_h b_\ell - a_\ell b_h < 0. \tag{A1}$$

Let $P' = \sum_{m=1}^{s} p_m$, $A' = \sum_{m=1}^{s} a_m p_m$ and $B' = \sum_{m=1}^{s} b_m p_m$. Multiplying both sides of (A1) by $p_\ell$ and $p_h$, and summing over $\ell = 1, \cdots, s$ and $h = s+1, \cdots, n+1$, we have that

$$
\begin{aligned}
0 &> \sum_{\ell=1}^{s} \sum_{h=s+1}^{n+1} \Big( b_h p_h p_\ell A - b_\ell p_\ell p_h A + a_\ell p_\ell p_h B - a_h p_h p_\ell B + a_h p_h b_\ell p_\ell - a_\ell p_\ell b_h p_h \Big) \\
&= (B - B')P'A - B'(1 - P')A + A'(1 - P')B - (A - A')P'B + (A - A')B' - A'(B - B') \\
&= 0,
\end{aligned}
$$

which is a contradiction. Therefore the result holds for $N = n + 1$. $\qquad \square$

To show Proposition 2, for each $t$ we construct $\tilde{\theta}_t^*$ that satisfies (i), (ii) and (iii). Notice that, for each $k = 1, \cdots, K$, there is

$$
\begin{aligned}
\mathbb{E}_{t-1}\Big[\mu(\alpha(\theta_t), \theta^*)\big|\theta_t \in \Theta_k\Big] &= \sum_{\theta \in \Theta_k} \mathbb{P}(\theta_t = \theta | \theta_t \in \Theta_k)\mathbb{E}_{t-1}\Big[\mu(\alpha(\theta), \theta^*)\big|\theta_t \in \Theta_k\Big] \\
&= \sum_{\theta \in \Theta_k} \mathbb{P}(\theta_t = \theta | \theta_t \in \Theta_k)\mathbb{E}_{t-1}\Big[\mu(\alpha(\theta), \theta^*)\Big], \quad\quad\text{(A2)}
\end{aligned}
$$

and

$$
\begin{aligned}
I_{t-1}\big(\psi; Y_{\alpha(\theta_t)}\big|\theta_t \in \Theta_k\big) &= \sum_{\theta \in \Theta_k} \mathbb{P}(\theta_t = \theta | \theta_t \in \Theta_k)I_{t-1}\big(\psi; Y_{\alpha(\theta)}\big|\theta_t \in \Theta_k\big) \\
&= \sum_{\theta \in \Theta_k} \mathbb{P}(\theta_t = \theta | \theta_t \in \Theta_k)I_{t-1}\big(\psi; Y_{\alpha(\theta)}\big), \quad\quad\text{(A3)}
\end{aligned}
$$

where we used the fact that $\theta_t$ is independent of $\theta^*$ and $\psi$.

According to Lemma 1, at stage $t$, for each $k = 1, \cdots, K$, there exists two parameters $\theta_1^{k,t}, \theta_2^{k,t} \in \Theta_k$ and $r_{k,t} \in [0,1]$, such that

$$
r_{k,t} \cdot \mathbb{E}_{t-1}\Big[\mu\big(\alpha(\theta_1^{k,t}), \theta^*\big)\Big] + (1 - r_{k,t}) \cdot \mathbb{E}_{t-1}\Big[\mu\big(\alpha(\theta_2^{k,t}), \theta^*\big)\Big] \leq \mathbb{E}_{t-1}\Big[\mu(\alpha(\theta_t), \theta^*)\big|\theta_t \in \Theta_k\Big], \tag{A4}
$$

and

$$
r_{k,t} \cdot I_{t-1}\Big(\psi; Y_{\alpha(\theta_1^{k,t})}\Big) + (1 - r_{k,t}) \cdot I_{t-1}\Big(\psi; Y_{\alpha(\theta_2^{k,t})}\Big) \leq I_{t-1}\big(\psi; Y_{\alpha(\theta_t)}\big|\theta_t \in \Theta_k\big). \tag{A5}
$$

Let $\tilde{\theta}_t^*$ be a random variable such that

$$
\mathbb{P}_{t-1}\Big(\tilde{\theta}_t^* = \theta_1^{k,t}\big|\psi = k\Big) = r_{k,t}, \quad \mathbb{P}_{t-1}\Big(\tilde{\theta}_t^* = \theta_2^{k,t}\big|\psi = k\Big) = 1 - r_{k,t}, \tag{A6}
$$

and let $\tilde{\theta}_t$ be an iid copy of $\tilde{\theta}_t^*$. Since the value of $\tilde{\theta}_t^*$ only depends on $\psi$, (i) is satisfied. Also we have that

$$
\begin{aligned}
I_{t-1}\Big(\psi; (\tilde{\theta}_t, Y_{\alpha(\tilde{\theta}_t)})\Big) &= I_{t-1}\Big(\psi; \tilde{\theta}_t\Big) + I_{t-1}\Big(\psi; Y_{\alpha(\tilde{\theta}_t)}\big|\tilde{\theta}_t\Big) \\
&\overset{(a)}{=} I_{t-1}\Big(\psi; Y_{\alpha(\tilde{\theta}_t)}\big|\tilde{\theta}_t\Big) \\
&= \sum_{k=1}^{K} \sum_{i=1,2} \mathbb{P}\big(\tilde{\theta}_t = \theta_i^{k,t}\big|\theta_t \in \Theta_k\big) \cdot \mathbb{P}\big(\theta_t \in \Theta_k\big) I_{t-1}\Big(\psi; Y_{\alpha(\theta_i^{k,t})}\Big) \\
&= \sum_{k=1}^{K} \Big[ r_{k,t} \cdot I_{t-1}\Big(\psi; Y_{\alpha(\theta_1^{k,t})}\Big) + (1 - r_{k,t}) \cdot I_{t-1}\Big(\psi; Y_{\alpha(\theta_2^{k,t})}\Big) \Big] \cdot \mathbb{P}\big(\theta_t \in \Theta_k\big) \\
&\overset{(b)}{\leq} I_{t-1}\big(\psi; Y_{\alpha(\theta_t)}\big|\theta_t \in \Theta_k\big) \cdot \mathbb{P}\big(\theta_t \in \Theta_k\big) \\
&= I_{t-1}\big(\psi; Y_{\alpha(\theta_t)}\big) \\
&\overset{(c)}{=} I_{t-1}\big(\psi; Y_{\alpha(\theta_t)}\big|\theta_t\big) + I_{t-1}\big(\psi; \theta_t\big) = I_{t-1}\big(\psi; (\theta_t, Y_{\alpha(\theta_t)})\big), \quad\quad\text{(A7)}
\end{aligned}
$$

where $(a)$ and $(c)$ follows from that both $\theta_t$ and $\tilde{\theta}_t$ are independent of $\psi$, conditioned on $\tilde{\mathcal{H}}_{t-1}$, and $(b)$ follows from (A5). Therefore (iii) is satisfied.

To show (ii), By construction we have that, at each stage $t = 1, \cdots, T$,

$$D_t = r_{k,t} \cdot \mathbb{E}_{t-1}\Big[\mu\big(\alpha(\theta_1^{k,t}), \theta^*\big)\Big] + (1 - r_{k,t}) \cdot \mathbb{E}_{t-1}\Big[\mu\big(\alpha(\theta_2^{k,t}), \theta^*\big)\Big] - \mathbb{E}_{t-1}\Big[\mu\big(\alpha(\theta_t), \theta^*\big)|\theta_t \in \Theta_k\Big] \leq 0.$$

Hence there is

$$
\begin{aligned}
\mathbb{E}_{t-1}\Big[R(Y_{\alpha(\tilde{\theta}_t)}) - R(Y_{\alpha(\theta_t)})\Big] &= \mathbb{E}_{t-1}\Big[\mu\big(\alpha(\tilde{\theta}_t), \theta^*\big) - \mu\big(\alpha(\theta_t), \theta^*\big)\Big] \\
&= \sum_{k=1}^{K} \mathbb{P}\Big(\theta_t \in \Theta_k\Big) \cdot \mathbb{E}_{t-1}\Big[\mu\big(\alpha(\tilde{\theta}_t), \theta^*\big) - \mu\big(\alpha(\theta_t), \theta^*\big) \mid \theta_t \in \Theta_k\Big] \\
&= \sum_{k=1}^{K} \mathbb{P}\Big(\theta_t \in \Theta_k\Big) \cdot D_t \leq 0. \tag{A8}
\end{aligned}
$$

Therefore we arrive at

$$
\begin{aligned}
&\mathbb{E}_{t-1}\Big[R^* - R(Y_{\alpha(\theta_t)})\Big] - \mathbb{E}_{t-1}\Big[R(Y_{\alpha(\tilde{\theta}_t^*)}) - R(Y_{\alpha(\tilde{\theta}_t)})\Big] \\
=\ &\mathbb{E}_{t-1}\Big[R(Y_{\alpha(\theta^*)}) - R(Y_{\alpha(\tilde{\theta}_t^*)})\Big] + \mathbb{E}_{t-1}\Big[R(Y_{\alpha(\tilde{\theta}_t)}) - R(Y_{\alpha(\theta_t)})\Big] \\
\leq\ &\mathbb{E}_{t-1}\Big[\mu(\alpha(\theta^*), \theta^*) - \mu(\alpha(\tilde{\theta}_t^*), \theta^*)\Big] \\
\leq\ &\epsilon, \tag{A9}
\end{aligned}
$$

where the final step comes from the fact that $\theta^*$ and $\tilde{\theta}_t^*$ are always in the same partition.

## B  Proof of Proposition 3

First, for two random parameters $\theta$ and $\theta'$ we define

$$\tilde{\Gamma}_t(\theta; \theta') = \frac{\mathbb{E}_{t-1}\big[R(Y_{\alpha(\theta)}) - R(Y_{\alpha(\theta')})\big]^2}{I_{t-1}\big(\theta; (\theta', Y_{\alpha(\theta')})\big)}, \tag{A10}$$

where the subscript $t - 1$ indicates the corresponding value under base measure $\tilde{\mathcal{H}}_{t-1}$. From the definition, $\tilde{\Gamma}_t(\theta; \theta')$ is a random variable measurable with respect to $\sigma(\tilde{\mathcal{H}}_{t-1})$.

**Lemma 2.** *We have that, for each $t = 1, \cdots, T$,*

$$
\begin{aligned}
I_{t-1}\Big(\tilde{\theta}_t^*; (\tilde{\theta}_t, Y_{\alpha(\tilde{\theta}_t)})\Big) &= \sum_{i=1}^{m} \mathbb{P}_{t-1}\Big(\tilde{\theta}_t = \theta^i\Big) I_{t-1}\Big(\tilde{\theta}_t^*; Y_{\alpha(\theta^i)}\Big) \\
&\geq 2 \sum_{i=1}^{m} \sum_{j=1}^{m} \mathbb{P}_{t-1}\Big(\tilde{\theta}_t^* = \theta^i\Big) \mathbb{P}_{t-1}\Big(\tilde{\theta}_t^* = \theta^j\Big) \cdot \\
&\quad \Big\{\mathbb{E}_{t-1}\big[R(Y_{\alpha(\theta^i)})|\tilde{\theta}_t^* = \theta^j\big] - \mathbb{E}_{t-1}\big[R(Y_{\alpha(\theta^i)})\big]\Big\}
\end{aligned}
$$

*and*

$$\mathbb{E}_{t-1}\big[R(Y_{\tilde{\theta}_t^*}) - R(Y_{\tilde{\theta}_t})\big] = \sum_{i=1}^{m} \mathbb{P}_{t-1}\Big(\tilde{\theta}_t^* = \theta^i\Big)\Big\{\mathbb{E}_{t-1}\big[R(Y_{\alpha(\theta^i)})|\tilde{\theta}_t^* = \theta^i\big] - \mathbb{E}_{t-1}\big[R(Y_{\alpha(\theta^i)})\big]\Big\},$$

*almost surely.*

**Proof.** For each $t = 1, \cdots, T$, there is

$$
\begin{aligned}
I_{t-1}\left(\tilde{\theta}_t^*; (\tilde{\theta}_t, Y_{\alpha(\tilde{\theta}_t)})\right) &= I_{t-1}\left(\tilde{\theta}_t^*; \tilde{\theta}_t\right) + I_{t-1}\left(\tilde{\theta}_t^*; Y_{\alpha(\tilde{\theta}_t)}|\tilde{\theta}_t\right) \\
&\overset{(d)}{=} I_{t-1}\left(\tilde{\theta}_t^*; Y_{\alpha(\tilde{\theta}_t)}|\tilde{\theta}_t\right) \\
&= \sum_{i=1}^m \mathbb{P}\left(\tilde{\theta}_t = \theta^i\right) \cdot I_{t-1}\left(\tilde{\theta}_t^*; Y_{\alpha(\tilde{\theta}_t)}|\tilde{\theta}_t = \theta^i\right) \\
&= \sum_{i=1}^m \mathbb{P}\left(\tilde{\theta}_t^* = \theta^i\right) \cdot I_{t-1}\left(\tilde{\theta}_t^*; Y_{\alpha(\theta^i)}\right) \\
&= \sum_{i=1}^m \sum_{j=1}^m \mathbb{P}\left(\tilde{\theta}_t^* = \theta^i\right)\mathbb{P}\left(\tilde{\theta}_t^* = \theta^j\right) \cdot D_{\mathrm{KL}}\left(P_{t-1}\left(Y_{\alpha(\theta^i)}|\tilde{\theta}_t^* = \theta^j\right)\big\| P_{t-1}\left(Y_{\alpha(\theta^i)}\right)\right) \\
&\overset{(e)}{\geq} 2\sum_{i=1}^m \sum_{j=1}^m \mathbb{P}\left(\tilde{\theta}_t^* = \theta^i\right)\mathbb{P}\left(\tilde{\theta}_t^* = \theta^j\right) \cdot \left\{\mathbb{E}_{t-1}\left[R(Y_{\alpha(\theta^i)})|\tilde{\theta}_t^* = \theta^j\right] - \mathbb{E}_{t-1}\left[R(Y_{\alpha(\theta^i)})\right]\right\}^2,
\end{aligned}
$$

where $(d)$ comes from the fact that $\tilde{\theta}_t^*$ and $\tilde{\theta}_t$ are independent, conditioned on $\mathcal{H}_{t-1}$, and $(e)$ follows from Pinsker's inequality and our assumption that $\sup_{y\in\mathcal{Y}} R(y) - \inf_{y\in\mathcal{Y}} R(y) \leq 1$.

On the other hand, there is also

$$
\begin{aligned}
\mathbb{E}_{t-1}\left[R(Y_{\tilde{\theta}_t^*}) - R(Y_{\tilde{\theta}_t})\right] &= \sum_{i=1}^m \mathbb{P}_{t-1}\left(\tilde{\theta}_t^* = \theta^i\right)\mathbb{E}_{t-1}\left[R(Y_{\alpha(\theta_i)})|\tilde{\theta}_t^* = \theta^i\right] - \\
&\quad \sum_{i=1}^m \mathbb{P}_{t-1}\left(\tilde{\theta}_t = \theta^i\right)\mathbb{E}_{t-1}\left[R(Y_{\alpha(\theta^i)})\right] \\
&= \sum_{i=1}^m \mathbb{P}_{t-1}\left(\tilde{\theta}_t^* = \theta^i\right)\left\{\mathbb{E}_{t-1}\left[R(Y_{\alpha(\theta^i)})|\tilde{\theta}_t^* = \theta^i\right] - \mathbb{E}_{t-1}\left[R(Y_{\alpha(\theta^i)})\right]\right\}.
\end{aligned}
$$

All equalities and inequalities hold almost surely. Thus the proof is complete. $\qquad\square$

**Lemma 3.** *For each $t = 1, 2, \cdots$, there is*

$$
\tilde{\Gamma}_t(\tilde{\theta}_t^*; \tilde{\theta}_t) \leq \frac{d}{2}, \quad \text{a.s.}
$$

**Proof.** Fix $t \in \{1, \cdots, T\}$, and let

$$
q_i = \mathbb{P}_{t-1}\left(\tilde{\theta}_t^* = \theta^i\right), \ s_i = \mathbb{E}_{t-1}\left[\theta^*|\tilde{\theta}_t^* = \theta^i\right], \quad i = 1, \cdots, m,
$$

and $s = \mathbb{E}_{t-1}\left[\theta^*\right]$. The linearity of expectation gives us

$$
\mathbb{E}_{t-1}\left[R(Y_{\alpha(\theta^i)})|\tilde{\theta}_t^* = \theta^j\right] = \alpha(\theta_i)^\top s_j, \quad \mathbb{E}_{t-1}\left[R(Y_{\alpha(\theta^i)})\right] = \alpha(\theta^i)^\top s, \quad \forall i,j \in \{1, \cdots, m\}.
$$

From Lemma 2, we have

$$
\begin{aligned}
\tilde{\Gamma}_t(\tilde{\theta}_t^*, \tilde{\theta}_t) &= \frac{\mathbb{E}_{t-1}\left[R(Y_{\alpha(\tilde{\theta}_t^*)}) - R(Y_{\alpha(\tilde{\theta}_t)})\right]^2}{I_{t-1}\left(\tilde{\theta}_t^*; (\tilde{\theta}_t, Y_{\alpha(\tilde{\theta}_t)})\right)} \\
&\leq \frac{\left(\sum_{i=1}^m q_i(\alpha(\theta^i)^\top s_i - \alpha(\theta^i)^\top s)\right)^2}{2\sum_{i=1}^m \sum_{j=1}^m q_i q_j(\alpha(\theta^i)^\top s_j - \alpha(\theta^i)^\top s)^2} \\
&= \frac{\left(\sum_{i=1}^m q_i \alpha(\theta_i)^\top (s_i - s)\right)^2}{2\sum_{i=1}^m \sum_{j=1}^m q_i q_j\left[\alpha(\theta_i)^\top (s_j - s)\right]^2} \quad \text{a.s.}
\end{aligned}
$$

Let $u_i = \sqrt{q_i}\alpha(\theta^i)$ and $v_i = \sqrt{q_i}(s_i - s)$, then $u_i, v_i \in \mathbb{R}^d$ for $i = 1, \cdots, m$. Consider the matrix

$$
M = (u_i^\top v_j)_{i,j=1}^m = \begin{pmatrix} u_1^\top \\ u_2^\top \\ \vdots \\ u_m^\top \end{pmatrix} \begin{pmatrix} v_1 & v_2 & \cdots & v_m \end{pmatrix}.
$$

Notice that $M$ is the product of an $m \times d$ matrix and a $d \times m$ matrix, hence $\mathrm{rank}(m) \leq d$. Therefore we have

$$\tilde{\Gamma}_t(\tilde{\theta}_t^*, \tilde{\theta}_t) \leq \frac{\mathrm{Trace}(M)^2}{2\|M\|_{\mathrm{F}}^2} \leq \frac{\mathrm{rank}(M)}{2} \leq \frac{d}{2}, \quad \text{a.s.}$$

$\square$

Notice that

$$
\begin{aligned}
\mathbb{E}\big[R(Y_{\alpha(\theta_1)}) - R(Y_{\alpha(\theta_2)})\big]^2 &\leq \mathbb{E}\Big[\mathbb{E}_{t-1}\big[R(Y_{\alpha(\theta_1)}) - R(Y_{\alpha(\theta_2)})\big]^2\Big] \\
&= \mathbb{E}\Big[\tilde{\Gamma}_t(\theta_1; \theta_2) \cdot I_{t-1}\big(\theta_1; (\theta_2, Y_{\alpha(\theta_2)})\big)\Big] \\
&\overset{(f)}{\leq} \frac{d}{2} \cdot \mathbb{E}\Big[I_{t-1}\big(\theta_1; (\theta_2, Y_{\alpha(\theta_2)})\big)\Big] \\
&= \frac{d}{2} \cdot I\big(\theta_1; (\theta_2, Y_{\alpha(\theta_2)})\big|\tilde{\mathcal{H}}_{t-1}\big), \quad\quad\quad\quad \text{(A11)}
\end{aligned}
$$

where $(f)$ comes from Lemma 3. Hence the proof is complete.

## C  Proof of Proposition 4

Let $\{\mathcal{A}_k\}_{k=1}^K$ be an $2\epsilon$-covering of $\mathcal{A}$ with respect to the Euclidean norm, i.e.
$$\|a_1 - a_2\|_2 \leq 2\epsilon, \quad \forall a_1, a_2 \in \mathcal{A}_k, \ k = 1, \cdots, K.$$

Define
$$\Theta_k = \alpha^{-1}(\mathcal{A}_k) = \{\theta \in \Theta : \alpha(\theta) \in \mathcal{A}_k\}.$$
Apparently $\{\Theta_k\}_{k=1}^K$ is a partition of $\Theta$. Moreover, for any $k \in \{1, \cdots, K\}$ and $\theta_1, \theta_2 \in \Theta_k$ there is

$$
\begin{aligned}
\big|\mu(\alpha(\theta_1), \theta_2) - \mu(\alpha(\theta_2), \theta_2)\big| &= \frac{1}{2}\big|\alpha(\theta_1)^\top \theta_2 - \alpha(\theta_2)^\top \theta_2\big| \\
&\leq \frac{1}{2}\|\alpha(\theta_1) - \alpha(\theta_2)\|_2 \cdot \|\theta_2\|_2 \leq \epsilon, \quad\quad \text{(A12)}
\end{aligned}
$$

where the last inequality follows from that $\|a_1 - a_2\|_2 \leq 2\epsilon$ and that $\Theta \subseteq \overline{\mathbf{B}_d(0,1)}$.

Let $N(S, \epsilon, \|\cdot\|)$ be the $\epsilon$-covering number of set $S$ with respect to the $\|\cdot\|$-norm. We only have to bound $N(\mathcal{A}, 2\epsilon, \|\cdot\|_2)$. From a standard result,

$$N(\mathcal{A}, 2\epsilon, \|\cdot\|_2) \leq N\left(\overline{\mathbf{B}_d(0,1)}, 2\epsilon, \|\cdot\|_2\right) \leq \left(\frac{1}{\epsilon} + 1\right)^d.$$

Therefore

$$K \leq \left(\frac{1}{\epsilon} + 1\right)^d.$$

## D  Proof of Theorem 2

From Theorem 1, Propositions 3 and 4, we have that for all $\epsilon > 0$,

$$\mathrm{BayesRegret}(T; \pi^{\mathrm{TS}}) \leq \sqrt{\frac{d}{2} \cdot d\log\left(\frac{1}{\epsilon} + 1\right) \cdot T} + \epsilon \cdot T.$$

Taking $\epsilon = d/\sqrt{2T}$, we arrive at

$$
\begin{aligned}
\mathrm{BayesRegret}(T; \pi^{\mathrm{TS}}) &\leq d\sqrt{\frac{T}{2}}\left(\sqrt{\log\left(1 + \frac{\sqrt{2T}}{d}\right)} + 1\right) \\
&\leq d\sqrt{T} \cdot \sqrt{\log\left(1 + \frac{\sqrt{2T}}{d}\right) + 1} \\
&\leq d\sqrt{T\log\left(3 + \frac{3\sqrt{2T}}{d}\right)}.
\end{aligned}
$$

# E Proof of Propositions 5, 6 and Theorem 3

Let $W$ be a random variable with the same distribution as the noise $W_a$ for all $a \in \mathcal{A}$. Define function $f$ as

$$f(x) = \mathbb{E}\big[\phi^{-1}(x - W)\big].$$

For each $a \in \mathcal{A}$, let $S_a = f(R_a) = \mathbb{E}\big[\phi^{-1}(R_a - W)\big|R_a\big]$. Then we have

$$
\begin{aligned}
\mathbb{E}\big[S_a\big|\theta^* = \theta\big] &= \mathbb{E}\Big[\mathbb{E}\big[\phi^{-1}(R_a - W)\big|R_a\big]\Big|\theta^* = \theta\Big] \\
&\overset{(g)}{=} \mathbb{E}\Big[\mathbb{E}\big[\phi^{-1}(R_a - W_a)\big|R_a\big]\Big|\theta^* = \theta\Big] \\
&\overset{(h)}{=} \mathbb{E}\Big[\mathbb{E}\big[a^\top \theta\big|R_a\big]\Big|\theta^* = \theta\Big] \\
&= a^\top \theta, \qquad\qquad\qquad\qquad\qquad\qquad\qquad\qquad (A13)
\end{aligned}
$$

where $(g)$ follows from the fact that $W$ and $W_a$ have the same distribution, and $(h)$ results from that conditioned on $p^* = p$,

$$R_a = \phi(a^\top \theta) + W_a.$$

From Lemma 3, we have that

$$\frac{\mathbb{E}_{t-1}\big[S_{\alpha(\tilde{\theta}_t^*)} - S_{\alpha(\tilde{\theta}_t)}\big]^2}{I_{t-1}\big(\tilde{\theta}_t^*; (\tilde{\theta}_t, S_{\alpha(\tilde{\theta}_t)})\big)} \leq 2d.$$

Notice that the constant is different from that in Lemma 3 since we have $S_a \in [-1, 1]$ for all $a \in \mathcal{A}$, whereas in Lemma 3 there is $R_a \in [-\nicefrac{1}{2}, \nicefrac{1}{2}]$. From data-processing inequality, since $S_{\alpha(\tilde{\theta}_t)} = f\big(R_{\alpha(\tilde{\theta}_t)}\big)$, there should be

$$I\big(\tilde{\theta}_t^*; (\tilde{\theta}_t, S_{\alpha(\tilde{\theta}_t)})\big) \leq I\big(\tilde{\theta}_t^*; (\tilde{\theta}_t, R_{\alpha(\tilde{\theta}_t)})\big).$$

Also there is

$$
\begin{aligned}
\mathbb{E}\big[S_{\alpha(\tilde{\theta}_t^*)} - S_{\alpha(\tilde{\theta}_t)}\big]^2 &= \mathbb{E}\big[f\big(R_{\alpha(\tilde{\theta}_t^*)}\big) - f\big(R_{\alpha(\tilde{\theta}_t)}\big)\big]^2 \\
&\geq \big[\inf_x f'(x)\big]^2 \cdot \mathbb{E}\big[R_{\alpha(\tilde{\theta}_t^*)} - R_{\alpha(\tilde{\theta}_t)}\big]^2 \\
&\overset{(i)}{\geq} C(\phi)^{-2} \cdot \mathbb{E}\big[R_{\alpha(\tilde{\theta}_t^*)} - R_{\alpha(\tilde{\theta}_t)}\big]^2,
\end{aligned}
$$

where $(i)$ is the consequence of

$$\inf_x f'(x) = \inf_x \mathbb{E}\big[(\phi^{-1})'(x - W)\big] \geq \inf_x (\phi^{-1})'(x) = \left[\sup_{y \in [-1,1]} \phi'(y)\right]^{-1}.$$

Therefore there is

$$\tilde{\Gamma}_t(\tilde{\theta}_t^*; \tilde{\theta}_t) \leq 2C(\phi)^2 d$$

where $\tilde{\Gamma}$ is defined in (A10). This proves Proposition 5.

On the other hand, let $\{\mathcal{A}_k\}_{k=1}^K$ be an $\epsilon/C(\phi)$-covering of $\mathcal{A}$ with respect to the Euclidean norm, i.e.

$$\|a_1 - a_2\|_2 \leq \epsilon/C(\phi), \quad \forall a_1, a_2 \in \mathcal{A}_k, \ k = 1, \cdots, K.$$

Define

$$\Theta_k = \alpha^{-1}(\mathcal{A}_k) = \{\theta \in \Theta : \alpha(\theta) \in \mathcal{A}_k\}.$$

Apparently $\{\Theta_k\}_{k=1}^K$ is a partition of $\Theta$. Moreover, for any $k \in \{1, \cdots, K\}$ and $\theta_1, \theta_2 \in \Theta_k$ there is

$$
\begin{aligned}
\big|\mu(\alpha(\theta_1), \theta_2) - \mu(\alpha(\theta_2), \theta_2)\big| &= \big|\phi(\alpha(\theta_1)^\top \theta_2) - \phi(\alpha(\theta_2)^\top \theta_2)\big| \\
&= C(\phi) \cdot \big|\alpha(\theta_1)^\top \theta_2 - \alpha(\theta_2)^\top \theta_2\big| \\
&\leq C(\phi) \cdot \|\alpha(\theta_1) - \alpha(\theta_2)\|_2 \cdot \|\theta_2\|_2 \leq \epsilon, \qquad (A14)
\end{aligned}
$$

where the last inequality follows from that $\|a_1 - a_2\|_2 \leq \epsilon/C(\phi)$ and that $\Theta \subseteq \overline{\mathbf{B}_d(0, 1)}$.

Similar as in the proof of Proposition 4,

$$N(\mathcal{A}, \epsilon/C(\phi), \|\cdot\|_2) \leq N\left(\overline{\mathbf{B}_d(0,1)}, \epsilon/C(\phi), \|\cdot\|_2\right) \leq \left(\frac{2C(\phi)}{\epsilon} + 1\right)^d.$$

Therefore

$$K \leq \left(\frac{2C(\phi)}{\epsilon} + 1\right)^d.$$

Therefore by choosing $\epsilon = \sqrt{2}C(\phi)d/\sqrt{T}$ in Theorem 1, we arrive at

$$
\begin{aligned}
\text{BayesRegret}(T; \pi^{\text{TS}}) &\leq \sqrt{2C(\phi)^2 d \cdot d \log\left(1 + \frac{\sqrt{2T}}{d}\right) \cdot T} + \sqrt{2}C(\phi)d\sqrt{T} \\
&\leq \sqrt{2}C(\phi)d\sqrt{T}\left(\sqrt{\log\left(1 + \frac{\sqrt{2T}}{d}\right)} + 1\right) \\
&\leq 2C(\phi)d\sqrt{T} \cdot \sqrt{\log\left(1 + \frac{\sqrt{2T}}{d}\right) + 1} \\
&\leq 2C(\phi)d\sqrt{T \log\left(3 + \frac{3\sqrt{2T}}{d}\right)}.
\end{aligned}
$$

## F   Proof of Theorem 4

For simplicity, we omit the superscript L in $\phi^{\text{L}}$ throughout this proof. We first show that, for any $\epsilon \in (0, \phi(\delta) - 1/2)$ there exists a partition $\{\Theta_k\}_{k=1}^K$ such that (3) holds and

$$K \leq \frac{1}{\epsilon}\left(1 + \frac{2}{\delta - \phi^{-1}(\phi(\delta) - \epsilon)}\right)^d. \tag{A15}$$

Let real-number sequence $s_0, s_1, \cdots, s_L$ be defined by

$$
\begin{aligned}
s_0 &= \phi^{-1}(\phi(\delta) - \epsilon), \\
s_1 &= \delta, \\
s_2 &= \phi^{-1}(\phi(\delta) + \epsilon), \\
s_3 &= \phi^{-1}(\phi(\delta) + 2\epsilon), \\
&\cdots \\
s_{L-1} &= \phi^{-1}(\phi(\delta) + (L-2)\epsilon), \\
s_L &= 1,
\end{aligned}
$$

where we choose $L$ such that $\phi(\delta) + (L-2)\epsilon < \phi(1) \leq \phi(\delta) + (L-1)\epsilon$. In addition, let $s'_j = -s_j$ for $j = 0, \ldots, L$. Notice that since $0 < \epsilon < \phi(\delta) - 1/2$, we have $s_0 > 0$. For $\ell = 1, \cdots, L-1$, let

$$\mathcal{Q}_\ell = \left\{\theta \in \Theta : s_\ell < \alpha(\theta)^\top \theta \leq s_{\ell+1}\right\},$$

and let

$$\mathcal{Q}_0 = \left\{\theta \in \Theta : s_0 \leq \alpha(\theta)^\top \theta \leq s_1\right\}.$$

Similarly for $\ell = 1, \cdots, L-1$, we can define

$$\mathcal{Q}'_\ell = \left\{\theta \in \Theta : s'_{\ell+1} < \alpha(\theta)^\top \theta \leq s'_\ell\right\},$$

and

$$\mathcal{Q}'_0 = \left\{\theta \in \Theta : s'_1 \leq \alpha(\theta)^\top \theta \leq s'_0\right\}.$$

From our assumption there is $\delta \leq |\alpha(\theta)^\top \theta| \leq 1$ for all $\theta \in \Theta$, hence

$$\left(\bigcup_{\ell=0}^{L-1} \mathcal{Q}_\ell\right) \cup \left(\bigcup_{\ell=0}^{L-1} \mathcal{Q}'_\ell\right) = \Theta.$$

For each $\ell = 1, \ldots, L$, let $\{\mathcal{A}_{\ell j}\}_{j=1}^{J_\ell}$ be an $(s_\ell - s_{\ell-1})$-covering of $\mathcal{A}$ with respect to the Euclidean norm, i.e. for each $j = 1, \cdots, J_\ell$,

$$\|a_1 - a_2\|_2 \leq s_\ell - s_{\ell-1}, \quad \forall a_1, a_2 \in \mathcal{A}_{\ell j}.$$

And let $\{\mathcal{A}'_{\ell j}\}_{j=1}^{J'_\ell}$ be an $(s'_{\ell-1} - s'_\ell)$-covering of $\mathcal{A}$ with respect to the Euclidean norm. Correspondingly, let $\{\Theta_{\ell j}\}_{j=1}^{J_\ell}$ be defined by

$$\Theta_{\ell j} = \{\theta \in \mathcal{Q}_\ell : \alpha(\theta) \in \mathcal{A}_{\ell j}\}.$$

Then $\{\Theta_{\ell j}\}_{j=1}^{J_\ell}$ is a partition of $\mathcal{Q}_\ell$, and for each $j = 1, \cdots, J_\ell$, let $\theta, \theta' \in \Theta_{\ell j}$, there is

$$\begin{aligned}
\mu(\alpha(\theta), \theta) - \mu(\alpha(\theta'), \theta) &= \phi\big(\alpha(\theta)^\top \theta\big) - \phi\big(\alpha(\theta')^\top \theta\big) \\
&\overset{(j)}{\leq} \phi\big(\alpha(\theta)^\top \theta\big) - \phi\big(\alpha(\theta)^\top \theta - (s_\ell - s_{\ell-1})\big) \\
&\overset{(k)}{\leq} \phi\big(s_\ell\big) - \phi\big(s_\ell - (s_\ell - s_{\ell-1})\big) = \epsilon,
\end{aligned}$$

where $(j)$ comes from that

$$\alpha(\theta)^\top \theta - \alpha(\theta')^\top \theta \leq \|\alpha(\theta) - \alpha(\theta')\|_2 \|\theta\|_2 \leq s_\ell - s_{\ell-1},$$

and $(k)$ follows from the fact that $\phi(x) - \phi\big(x - (s_\ell - s_{\ell-1})\big)$ is decreasing in $x$ when $x > s_\ell - s_{\ell-1}$. Let $\{\Theta'_{\ell j}\}_{\ell,j}$ be the counterpart of $\{\Theta_{\ell j}\}_{\ell,j}$ defined with respect to $\{\mathcal{A}'_{\ell j}\}_{\ell,j}$, then $\{\Theta_{\ell j}\}_{\ell,j} \cup \{\Theta'_{\ell j}\}_{\ell,j}$ is a valid partition of $\Theta$. Notice that

$$s_1 - s_0 < s_2 - s_1 < \cdots < s_L - s_{L-1},$$

we thence have

$$\begin{aligned}
K &\leq \sum_{\ell=1}^{L} J_\ell + \sum_{\ell=1}^{L} J'_\ell \\
&\leq 2 \sum_{\ell=1}^{L} N(\mathcal{A}, s_\ell - s_{\ell-1}, \|\cdot\|_2) \\
&\leq 2L \cdot N(\mathcal{A}, s_1 - s_0, \|\cdot\|_2) \\
&\leq \frac{1}{\epsilon} \left(1 + \frac{2}{\delta - \phi^{-1}(\phi(\delta) - \epsilon)}\right)^d.
\end{aligned} \qquad (A16)$$

Hence we have proved (A15). Let $\Phi(x) = \delta - \phi^{-1}(\phi(\delta) - x)$, then there is $\Phi(0) = 0$ and $\Phi'(0) = \frac{1}{\phi(\delta)(1-\phi(\delta))}$. Also notice that

$$\begin{aligned}
\Phi''(0) &= -(\phi^{-1})''(\phi(\delta) - x)\Big|_{x=0} \\
&= \frac{2\phi(\delta) - 1}{(\phi(\delta) - \phi(\delta)^2)^2} > 0,
\end{aligned} \qquad (A17)$$

where we used the fact that $\phi(\delta) > 1/2$. Hence for small enough $\epsilon$, there is $\Phi(\epsilon) \geq \Phi'(0) \cdot \epsilon$. Notice that from Theorem 1 and Conjecture 1, for all $T$ and $\epsilon > 0$ there is

$$\begin{aligned}
\text{BayesRegret}(T; \pi^{\text{TS}}) &\leq \sqrt{\frac{d}{2} \cdot \log K \cdot T} + \epsilon \cdot T \\
&\leq \sqrt{\frac{d}{2} \cdot \left(-\log(\epsilon) + d \log\left(1 + \frac{2}{\Phi(\epsilon)}\right)\right) \cdot T} + \epsilon \cdot T \quad (A18)
\end{aligned}$$

Let $\epsilon = d/\sqrt{2T}$, for large enough $T$ we have

$$
\begin{aligned}
\text{BayesRegret}(T; \pi^{\text{TS}}) &\leq \sqrt{\frac{d}{2} \cdot \left( \log(\frac{\sqrt{2T}}{d}) + d \log \left( 1 + \frac{2\sqrt{2T}}{\Phi'(0)d} \right) \right) \cdot T} + d\sqrt{\frac{T}{2}} \\
&\leq \sqrt{\frac{d(d+1)}{2} \cdot T} \cdot \left( \sqrt{\log \left( 1 + \frac{2\sqrt{2T}}{\Phi'(0)d} \right)} + 1 \right) \\
&\leq \sqrt{d(d+1)T} \cdot \sqrt{\log \left( 1 + \frac{2\sqrt{2T}}{\Phi'(0)d} \right) + 1} \\
&\leq 2d\sqrt{T} \cdot \sqrt{\log \left( 3 + \frac{6\sqrt{2T}}{d} \cdot \frac{\beta e^{\beta\delta}}{(1 + e^{\beta\delta})^2} \right)}. \qquad\qquad \text{(A19)}
\end{aligned}
$$