[Reviews · NeurIPS 2018]

Reviewer 1



Using a simple rate-distortion argument, authors devise an information-theoretic analysis of Thompson sampling on arbitrarily large action spaces. they establish regret bound for Thompson sampling on both linear and generalized linear bandit problems. The proofs are clear and easy to follow however, this analysis relies on a conjecture about the information ratio of Thompson sampling for the logistic bandit, which is only supported through computational results. Since the paper is only about theory it makes this missing part important to conclude the proof. I have the following comment Problem of notation line 95 line. Could you please clarify the following, in the proof of theorem 2 (appendix ) line 78, how you could take \epsilon=d/sqrt(sT), and we know that \epsilon is constant ?

Reviewer 2



POST REBUTTAL: Thanks for the response. My scores stay the same. Of course I encourage the authors to try and relax the assumptions for the generalized linear models and prove the conjecture. Summary. The paper provides a Bayesian regret-analysis of Thompson sampling for linear bandits with large action spaces. Previous bounds depended on the number of actions, which is known to be suboptimal when the number of actions is super-exponential in the dimension. Existing Bayesian analysis for this case relied on the conversion from frequentist bounds, which is (a) excessively complicated and (b) leads to looser bounds. The authors give a simple and rather intuitive analysis based on the observation that the learner only needs to learn the unknown parameter up to some epsilon and so the entropy of the unknown parameter should be replaced by the entropy of a partitioning random variable. The new analysis is applied to the standard linear bandit model where O(d sqrt(T log(T))) bounds are provided that improve on the state-of-the-art by a factor of sqrt(log(T)). The bounds are also applied to generalized linear bandits. More comments on this to follow. Major comments. I think this is a nice paper. Some questions/comments. First, there is an assumption that the rewards are bounded and this seems quite important for the proofs. For the standard frequentist analysis, however, one assumes the rewards are subgaussian. Of course you can apply a union bound over all the observed rewards, but I think a log(T) factor in the regret will sneak in if you do this. Do you know how to relax the assumption on the noise to (say) conditionally subgaussian? Assumptions 1 and 3 both seem quite unreasonable. In applications of GLMs that I know of the rewards are almost always Bernoulli and the expected rewards are almost always very small. Eg., the probability that a user clicks on something. Can you say more about whether or not these assumptions can be relaxed? Clarity. The paper is mostly well written. I think some discussion of the assumptions is appropriate and there are a few typos (see below). Quality. The proofs seem to be correct. The problem is interesting. Originality. This is the weakest part of the paper. The idea to use a discretization is quite an obvious one and the analysis does not change that much from previous work in this area. Significance. I think there will be plenty of interest in this paper. The paper would be more significant if the assumptions for generalized linear models were more realistic. Overall. A good paper. Perhaps the authors 'stopped early' because there are lots of unresolved questions and the boat is not pushed too much further. Still, I prefer to see it accepted. Minors. * Is \mathcal Y finite? Measurability of g? Almost sure qualifications on conditional expectations. * Why not mention in (1) that I(. ; . | . ) is the conditional mutual information? * Both H_t and ~H_t generate different sigma-algebras, but the extra information in ~H_t does not change the conditional measure (which is just the posterior - though this is not unique... measurability, brr). Anyways, is there some subtle reason behind this change or do you just want to emphasize the view of sampling thetas? * In (d) of Eq. (6) should ~H_T be ~H_{T-1}? * L137. p^* should be theta^*? * You could be consistent about the order of theta^* and psi in (6) and the display after, though of course it is symmetric. * L158. \in -> \subset * L160. Did you mean to say that the action set and parameter space or dual to each other? Otherwise I don't see where you actually assume the inner product is bounded. The 'normalizing factor' of 1/2 is nonstandard. Why not just let mu(a, theta) = and assume || <= 1/2 for all a, theta? * Assumption 1 is quite restrictive. One reason to introduce GLMs in the first place is to behave well with Bernoulli noise. * Assumption 3 is also quite restrictive.

Reviewer 3



The paper devises a new technique for information theoretic analysis of Thompson Sampling. The main result is to recover the existing bounds for linear bandits, and provide new results for logistic bandits. The new technique is conceptually interesting. I have two main concerns. First, in terms of a general technique, Theorem 1 does not seem any more easy to use than the results in Russo and Van Roy relating regret to entropy. Also, the regret bounds for linear and generalized bandits are not a significant improvement over existing results. The results for logistic bandits are an improvement but depend on a conjecture. Overall, the paper is conceptually interesting and well written, but seems incremental over Russo and Van Roy. Post author-feedback: I thank the authors for the clarifying discussion. I went through it carefully, but am still not convinced with the significance of the new techniques. My score remains unchanged.